# Entropy engineering promotes thermoelectric performance in p-type chalcogenides

Binbin Jiang [1,5], Yong Yu[1,5], Hongyi Chen[2], Juan Cui[1], Xixi Liu[1], Lin Xie[1] & Jiaqing He [1,3,4 ✉]

We demonstrate that the thermoelectric properties of p-type chalcogenides can be effectively improved by band convergence and hierarchical structure based on a high-entropy-stabilized matrix. The band convergence is due to the decreased light and heavy band energy offsets by alloying Cd for an enhanced Seebeck coefficient and electric transport property. Moreover, the hierarchical structure manipulated by entropy engineering introduces all-scale scattering sources for heat-carrying phonons resulting in a very low lattice thermal conductivity. Consequently, a peak $zT$ of 2.0 at 900 K for p-type chalcogenides and a high experimental conversion efficiency of 12% at $\Delta T = 506$ K for the fabricated segmented modules are achieved. This work provides an entropy strategy to form all-scale hierarchical structures employing high-entropy-stabilized matrix. This work will promote real applications of low-cost thermoelectric materials.

---

[1] Shenzhen Key Laboratory of Thermoelectric Materials, Department of Physics, Southern University of Science and Technology, Shenzhen, China. [2] College of Chemistry and Chemical Engineering, Central South University, Changsha, China. [3] Guangdong-Hong Kong-Macao Joint Laboratory for Photonic-Thermal-Electrical Energy Materials and Devices, Southern University of Science and Technology, Shenzhen, China. [4] Key Laboratory of Energy Conversion and Storage Technologies, Southern University of Science and Technology, Ministry of Education, Shenzhen, China. [5]These authors contributed equally: B. Jiang, Y. Yu. ✉email: hejq@sustech.edu.cn

Thermoelectric (TE) technologies can generate electricity from waste heat and have drawn widespread attention because of the possibility of increasing overall energy efficiency[1–5]. However, its high cost and low conversion efficiency weaken the competitiveness[6–8]. To achieve a high conversion efficiency, TE materials with a high figure-of-merit ($zT$) should be used[9,10]:

$$\eta_{max} = \frac{T_h - T_c}{T_h} \frac{\sqrt{1 + z\frac{T_h+T_c}{2}} - 1}{\sqrt{1 + z\frac{T_h+T_c}{2}} + \frac{T_c}{T_h}},$$

(1)

where $T_h$ and $T_c$ are the hot- and cold-side temperatures. The dimensionless figure-of-merit $zT = S^2\sigma/\kappa$ is given by the material's parameters. Here, the Seebeck coefficient is $S$, the electrical conductivity is $\sigma$, and the thermal conductivity is $\kappa$. Many strategies have been proposed to improve $zT$ values, which aimed to increase $\sigma$ or $S$ or decrease $\kappa$. For example, decreasing carrier effective mass and scattering is beneficial to increase $\sigma$ because of the increased carrier mobility[11,12]. Band convergence and resonant level increase the density-of-states' effective mass contributing to a large $S$ (refs. [13,14]). Lattice defects and nanostructures destroy the propagation path of heat-carrying phonons resulting in a low $\kappa$ (refs. [15–17]).

Recently, entropy engineering has been used as a novel strategy to optimize the electrical and thermal transport properties of TE materials[18–20]. By increasing element species, the mixing entropy in a material system increases quickly, and high-entropy materials with over five principal elements will be formed[21]. Severe lattice distortion is usually the core phenomenon of high-entropy material—such distortion leads to strong scattering for heat-carrying phonons and results in low lattice thermal conductivity[18]. Stabilizing single-phase structures by increasing entropy weakens the phase-boundary electron scattering in multiphase high-entropy material, thus providing a beneficial contribution to improve electrical transport properties, i.e., entropy-driven structural stabilization[18,22]. Therefore, the high-entropy-stabilized composition should work as a good matrix for further optimization of TE performance by integrating the traditional optimization strategy.

PbSe is a promising alternative to expensive PbTe because of the earth-abundant element Se. Recently, the $zT$ value of n-type PbSe has been largely improved to a high value of 1.8, which is comparable to n-type PbTe[18,23,24]. However, p-type PbSe still shows a much lower $zT$ value than PbTe, which prevents its possible commercial applications[25–35]. Here, we synthesized high-entropy $Pb_{0.975}Na_{0.025}Se_{0.5}S_{0.25}Te_{0.25}$ compositions whose structure was stabilized by increasing the entropy. This increased the peak $zT$ to a high value of 2.0 at 900 K with band convergence and hierarchical structures (Fig. 1a). The hierarchical structures manipulated by introducing complicated elemental composition demonstrated different scales of lattice defects (Fig. 1c). These scatter all-frequency phonons and contributed to a very low lattice thermal conductivity ($\kappa_L$) of 0.33 Wm$^{-1}$K$^{-1}$ for $Pb_{0.935}Na_{0.025}Cd_{0.04}Se_{0.5}S_{0.25}Te_{0.25}$ sample. Based on the high-performance high-entropy chalcogenides, we fabricated a segmented TE module and realized an experimental conversion efficiency of 12% at $\Delta T = 506$ K, which is one of the highest results yet reported (Fig. 1b).

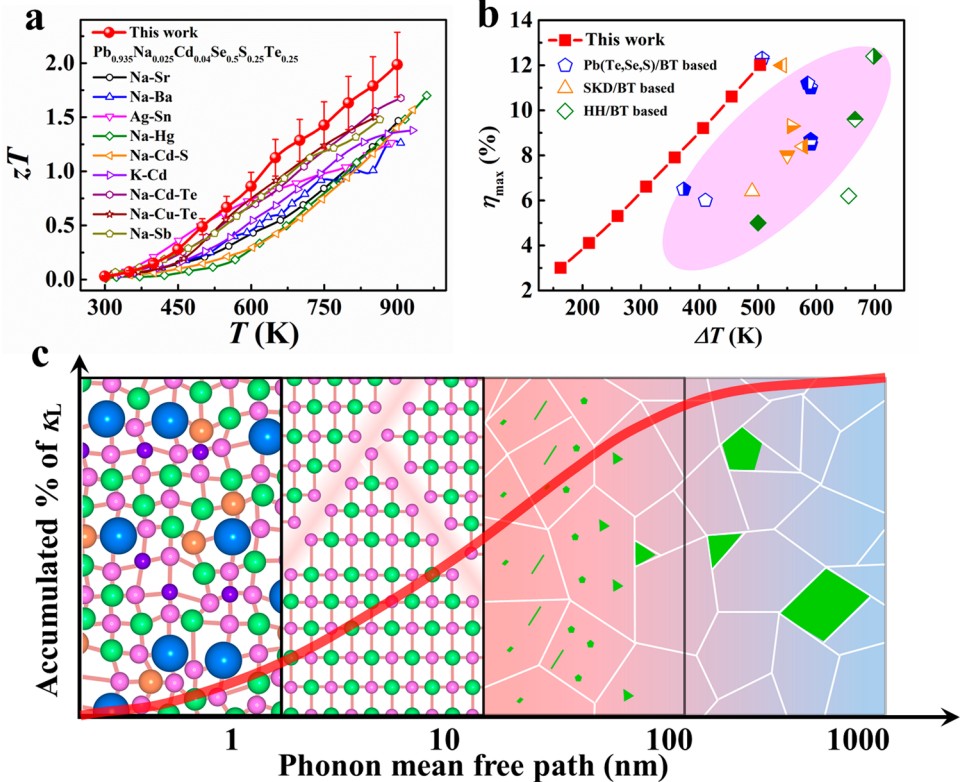

**Fig. 1 Improving TE performance in all-scale hierarchical structures through entropy engineering. a** Temperature dependence of $zT$ values for p-type chalcogenides in this work. The results of p-type PbSe-based materials in the literature are also included[25–35]. **b** Temperature difference ($\Delta T$) dependence of maximum conversion efficiencies ($\eta_{max}$) for chalcogenides-based segmented TE modules in this work. The results of Pb(Te,Se,S)/Bi$_2$Te$_3$-based[18,46,47,49–51], Skutterudite/Bi$_2$Te$_3$-based[48,52–56], and half-Heusler/Bi$_2$Te$_3$-based[57–59] TE modules in literature are also included (the pink area). **c** Phonon mean free path dependence of accumulated lattice thermal conductivity $\kappa_L$ (the red line) and schematic diagram of all-scale hierarchical structures (point defects, planar defects, nanoprecipitates, and mesoscopic precipitates). The pink, orange, green, blue, and purple spheres represent Pb, Na, Se, Te, and S atoms, respectively. The green and red grains represent CdS and high-entropy matrix, respectively.

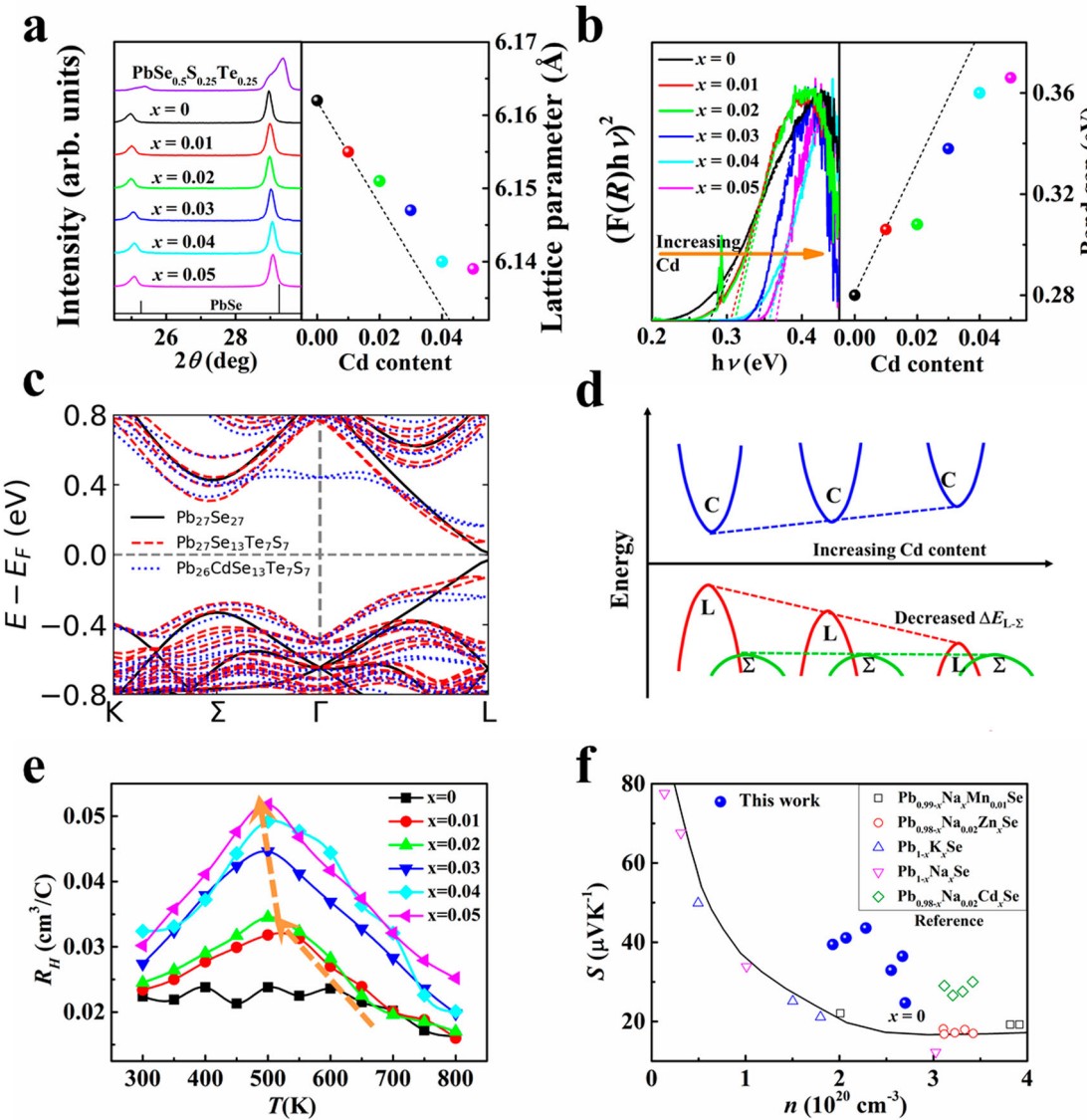

**Fig. 2 Band convergence in p-type Cd-doped chalcogenides. a** Powder XRD patterns (left), calculated lattice parameters (right), **b** room-temperature infrared diffuse reflectance spectroscopy (left) and calculated band gaps (right) for $Pb_{0.975-x}Cd_xNa_{0.025}Se_{0.5}S_{0.25}Te_{0.25}$ ($x = 0$, 0.01, 0.02, 0.03, 0.04, 0.05) samples. The spheres with different colors correspond to the lines with the same colors. **c** Calculated band structures of $Pb_{27}Se_{27}$, $Pb_{27}Se_{13}Te_7S_7$, and $Pb_{26}CdSe_{13}Te_7S_7$. **d** Schematic diagram of band convergence by alloying S, Te, and Cd in PbSe. **e** Temperature dependence of Hall coefficients (the orange arrows represent the decreased peak temperature), and **f** carrier concentration dependence of the Seebeck coefficients (the Pisarenko relationship) for $Pb_{0.975-x}Cd_xNa_{0.025}Se_{0.5}S_{0.25}Te_{0.25}$ ($x = 0$, 0.01, 0.02, 0.03, 0.04, 0.05) samples (blue spheres). The black line and open symbols are from reference[27].

## Results

**High-entropy composition and band structure**. Powder X-ray diffraction patterns at room temperature for all the samples are shown in Fig. 2a. The results reveal that the $PbSe_{0.5}S_{0.25}Te_{0.25}$ sample has split diffraction peaks, which come from the multi-phase mixture with modulated composition. Upon introducing Na, the split peaks converged into one peak illustrating the resulting single phase. This phenomenon was also observed in Sn-alloyed (Pb,Sn)(Se,S,Te) system and can be well explained by the entropy-driven structural stabilization[18,22]. The introduced Na increased the mixing entropy resulting in a negative Gibbs free energy and stabilized structure. The stabilized structure can keep the long-range order of atomic arrangement, thereby eliminating the boundary phonon scattering around phase interfaces[18]. This phenomenon of stabilized structure from entropy-driven structural stabilization can maintain the electrical transport framework

and improve the electrical properties[18], which is similar to the improved atomic ordering in Cd-doped $AgSbTe_2$ (ref. [36]). Meanwhile, the increased solubility of Te and S at anion sites from the increased entropy also extends phase space for performance optimization and largely distorts the lattice, resulting in strong scattering for heat-carrying phonons. Based on this single-phase composition ($Pb_{0.975}Na_{0.025}Se_{0.5}S_{0.25}Te_{0.25}$), we further alloyed Cd at the Pb site to tune the band structure and micro-structure. The sharp peaks of the $Pb_{0.975-x}Na_{0.025}Cd_x$-$Se_{0.5}S_{0.25}Te_{0.25}$ sample can be well indexed to rock salt PbSe structure, which demonstrated that the single-phase matrix was maintained upon introducing Cd.

The alloyed Cd should change the band structure because of the changed orbital hybridization[31] and the compressed lattice parameter (Fig. 2a), as proved by the diffuse reflectance spectra[27]. Based on Kubelka–Munk relationship, the band gaps of all of the

samples with different Cd contents were measured as shown in Fig. 2b. The band gap increased from 0.28 to 0.37 eV, when the Cd content increased from 0 to 0.05. The band structure based on density functional theory (DFT) was calculated to determine the effect of the increased band gap on band structure. As shown in Fig. 2c, the valence band maximum was pushed down, and the conduction band minimum was pushed up because of the increased band gap with introducing Cd; the other band valleys remained unchanged. As a result, the energy difference ($\Delta E_{L-\Sigma}$) between the light (L) and heavy ($\Sigma$) valence bands will be decreased, as shown in the schematic diagram in Fig. 2d. For p-type chalcogenides without alloying Cd, the Fermi level only pass through L band because of the large $\Delta E_{L-\Sigma}$. But the Fermi level in Cd-doped materials should be deep into both the L and $\Sigma$ bands due to the decreased $\Delta E_{L-\Sigma}$, which leads to the increased valley degeneracy $N_v$ (band convergence)[27,28]. PbSe is well known as a typical incipient metal with a unique bonding mechanism called metavalent bonding (MVB)[37]. MVB is mainly formed by p-orbitals in a $\sigma$-bonding configuration in chalcogenides, which shows high electron mobility because of the small conductivity effective mass and weak s–p hybridization[38]. In our experiment, alloying Cd at Pb site will strengthen s–p hybridization between cation and anion because of the increased charge sharing[37,39]. Thus, the band gap opens and band effective mass of a single valley ($m_b^*$) increases, resulting in the reduced charge carrier mobility (Fig. S1d)[40]. In this regard, alloying Cd should deteriorate the electrical transport property based on a single parabolic band model. However, the enlarged band gap decreases the energy separation between L and $\Sigma$ bands, and promotes the band convergence as verified by our DFT calculations. The participation of $\Sigma$ band in electrical transport process leads to the multiple band behavior[13]. As a consequence, the valley degeneracy $N_v$ and density-of-states effective mass $m^*$ ($m^* = N_v^{2/3} m_b^*$) will be largely increased (the Pisarenko line in Fig. 2f)[13], resulting in the enhanced S and power factor ($PF = S^2\sigma$).

**TE properties of high-entropy materials.** The changed band structure from the band convergence is largely tuned along the electrical transport properties. As shown in Fig. 3a, the electrical conductivity ($\sigma$) decreased and the Seebeck coefficient (S) increased with increasing Cd content due to the decreased carrier concentration (Fig. S1a). There was an obvious plateau in the temperature-dependent curves of S for the Cd-alloyed samples, which showed enhancement of S.

We performed the Hall measurement to analyze the origin of the increased S (Fig. 2e and Fig. S1). The Hall coefficient had an increased peak with increasing Cd content ~500 K (Fig. 2e), illustrating that the hole transport changed from single- to two-band behavior[28]. Generally, the Hall coefficient is expected to be temperature independent when the Fermi level only passes through single band. However, the monotonous trend turns into complex behavior because of the redistribution of holes in the two bands[29–31]. The participated contribution of both heavy and light bands to the hole transport process increased the band degeneracy ($N_v$) and overall density-of-states resulting in the enhanced S[13,28–31]. Therefore, the peak $PF$ was largely increased to 16.5 µWcm$^{-1}$K$^{-2}$ for the Pb$_{0.955}$Na$_{0.025}$Cd$_{0.02}$Se$_{0.5}$S$_{0.25}$Te$_{0.25}$ sample, which is 22% higher than that of Pb$_{0.975}$Na$_{0.025}$-Se$_{0.5}$S$_{0.25}$Te$_{0.25}$ sample (Fig. 3b).

Entropy engineering is an effective strategy to tune lattice thermal conductivity ($\kappa_L$)[18–20,41–43], which also worked in this system. The total thermal conductivity ($\kappa$) largely decreased from 5 to 2 Wm$^{-1}$K$^{-1}$ when the composition changed from low-entropy Pb$_{0.975}$Na$_{0.025}$Se to high-entropy Pb$_{0.975}$Na$_{0.025}$Se$_{0.5}$S$_{0.25}$Te$_{0.25}$ (Fig. 3c). By subtracting the contribution of the electronic thermal

conductivity ($\kappa_e$) based on the Wiedemann–Franz law[2,3], term $\kappa_L$ was calculated and plotted in Fig. 3d. Clearly, $\kappa_L$ is largely depressed to a low value in the entire temperature range for the high-entropy Pb$_{0.975}$Na$_{0.025}$Se$_{0.5}$S$_{0.25}$Te$_{0.25}$ sample, which was even much lower than that of the nanostructure PbSe[27,29,30,34], such as the green line for Pb$_{0.935}$Na$_{0.025}$Cd$_{0.04}$Se in this work and the reported Pb$_{0.95}$Na$_{0.02}$Cd$_{0.03}$Se in previous literature[27,34]. Furthermore, we introduced Cd to produce nanoprecipitates using this high-entropy-stabilized composition as a matrix material, resulting in a very low $\kappa_L$ in the whole temperature range. This temperature-independent trend of $\kappa_L$ largely deviates from the $T^{-1}$ relation for Umklapp phonon scattering[3,5,7], demonstrating that the scattering from a high-entropy matrix and nanostructure dominates the phonon transport process. The low $\kappa_L$ (0.33 Wm$^{-1}$K$^{-1}$) is close to the theoretical minimum value (0.31 Wm$^{-1}$K$^{-1}$) based on Cahill model[44], illustrating that the phonon mean free path is very small and similar to the atomic distance. The full-spectrum phonons should be strongly scattered by wide-scale scattering sources considering the temperature-independent trend of $\kappa_L$ and atomic-scale phonon mean free path[45].

The highest $zT$ value was largely improved to 2.0 at 900 K (Fig. 3e) by combining the improved $PF$ from band convergence and the depressed $\kappa_L$ from full-spectrum phonon scattering presented above. This is a high value in p-type PbSe-based materials and comparable to expensive p-type PbTe[13,45–47]. The $zT$ value of high-entropy Pb$_{0.935}$Na$_{0.025}$Cd$_{0.04}$Se$_{0.5}$S$_{0.25}$Te$_{0.25}$ sample has been improved by 100% and 67%, respectively, versus Pb$_{0.975}$Na$_{0.025}$Se and Pb$_{0.935}$Na$_{0.025}$Cd$_{0.04}$Se with only band convergence and nanostructure.

**TE module.** Although discussions in the TE community often focuses on the development of $zT$ value, the conversion efficiency of the module is more valuable to evaluate the the TE performance[18,46,47]. In this work, we fabricated a segmented module based on this p-type Pb$_{0.935}$Na$_{0.025}$Cd$_{0.04}$Se$_{0.5}$S$_{0.25}$Te$_{0.25}$ and n-type Pb$_{0.89}$Sb$_{0.012}$Sn$_{0.1}$Se$_{0.5}$S$_{0.25}$Te$_{0.25}$ as in our previous work[18] along with commercial p-type Bi$_{1.5}$Sb$_{0.5}$Te$_3$ and n-type Bi$_2$Te$_{2.7}$Se$_{0.3}$[18]. This eight-couple module had a dimension of 20 mm by 20 mm by 13.5 mm (inset in Fig. 3f). The output properties of the fabricated module were measured under different operating temperatures by PEM-2 (Fig. S2). Based on the measured voltage ($U$) and current ($I$), the output power ($P$) can be calculated as $P = UI$. A maximum $P$ of 2.7 W was obtained under a temperature difference $\Delta T = 506$ K ($\Delta T = T_h - T_c$. $T_h$, the temperature at hot side; $T_c$, the temperature at cold side; Fig. S2c). The conversion efficiency can be obtained as $\eta = P/(P + Q_c)$ (Fig. 3f) by measuring the heat flow at the cold side ($Q_c$, Fig. S2d)[18,48]. The measured $\eta$ reached a maximum value of 12% under $\Delta T = 506$ K, which is among the reported highest values for all the TE systems[18,46–59]. In this experiment, the height ratio of PbSe- and Bi$_2$Te$_3$-based legs ($H_{PbSe}/H_{BT}$) was set at 1.5 and the cross-sectional area ratio of the p- and n-legs ($A_p/A_n$) was set at 1.0, which were not fully optimized. Therefore, further improvements can be realized by optimizing the leg geometry factors. Meanwhile, the electrical and thermal losses at various interfaces (electrical and thermal contact resistance, thermal radiation and convection) will also degrade the conversion efficiency, which demand optimization for the assembly approach of the TE module[48].

**All-scale hierarchical structures.** The very low lattice thermal conductivity from atomic-scale phonon mean free path is the main contribution to the largely improved $zT$ value and conversion efficiency in this work. To analyze the origin of phonon scattering, we performed transmission electron microscopy (TEM) studies on the high-performance Pb$_{0.935}$Na$_{0.025}$Cd$_{0.04}$-Se$_{0.5}$S$_{0.25}$Te$_{0.25}$ sample (Fig. 4 and Figs. S3–S5). As shown in

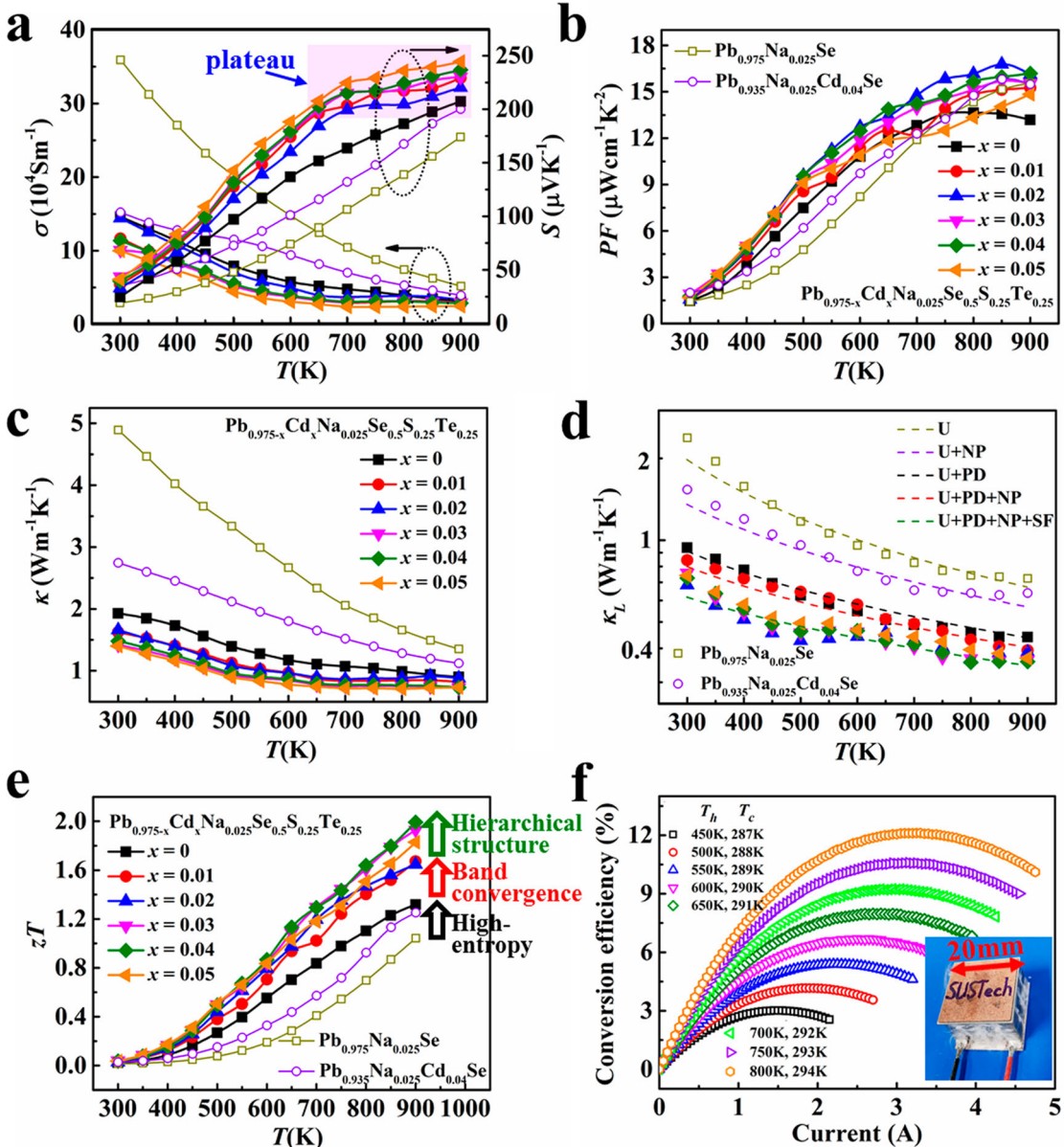

**Fig. 3 TE performance of chalcogenide-based materials and modules.** Temperature dependences of **a** electrical conductivity $\sigma$ (the left black arrow), the Seebeck coefficient $S$ (the right black arrow), **b** power factor $PF$, **c** total thermal conductivity $\kappa$, **d** lattice thermal conductivity $\kappa_L$, and **e** $zT$ values of $Pb_{0.975}Na_{0.025}Se$, $Pb_{0.935}Na_{0.025}Cd_{0.04}Se$, and $Pb_{0.975-x}Cd_xNa_{0.025}Se_{0.5}S_{0.25}Te_{0.25}$ ($x = 0$, 0.01, 0.02, 0.03, 0.04, 0.05) samples. The dash lines in **e** were calculated by the modified Debye–Callaway model. U, NP, PD, and SF represent Umklapp process, nanoprecipitates, point defects, and stacking faults scattering for phonons, respectively. **f** Current dependence of conversion efficiency under different operating temperatures ($T_h$, temperature at hot side; $T_c$, temperature at cold side) for the fabricated segmented TE module (the inset photograph).

Fig. 4a, high-density nanoprecipitates can be observed throughout the sample, whose size is distributed in a large range from several nanometers to several hundred nanometers. We then used energy-dispersive spectroscopy and selected area electron diffraction (SAED) to study the elemental composition and crystal structure of the nanoprecipitates. Figures S3 and S4 show that the nanoprecipitates are enriched with Cd and S, which is consistent with their dark contrast. Generally, CdS exhibits covalent bonding, which shows different physical properties, such as optical dielectric constant, effective coordination numbers, and Born effective charge compared with MVB materials Pb(Se,S,Te)[37–39]. Thus, there should be large phonon scattering at the MVB/covalent heterointerfaces due to the very different spring constant of chemical bonds, contributing to the ultralow $\kappa_L$ (ref. [60]). By

indexing diffraction spots in Fig. 4b, c, we found that the nanoprecipitates showed the same rock salt (face-centered cubic) structure as the PbSe-based matrix. The difference between the two phases was the lattice constant, which is inversely proportional to the reciprocal-space distance of spots in SAED patterns. The interfaces between the nanoprecipitates and matrix were semi-coherent (Fig. 4d), resulting in the periodic stacking faults due to the same crystal structure, but different lattice constants. The high-density stacking faults within the nanoprecipitates resulted in dense lattice strains, which were proved by the geometric phase analysis (GPA) as the insets in Fig. 4d–f show. In traditional nanocomposites, the lattice strains usually exist around the phase interfaces rather than within the grains[15,17,27,45,53,61,62]. Therefore, the size of the nanoprecipitates

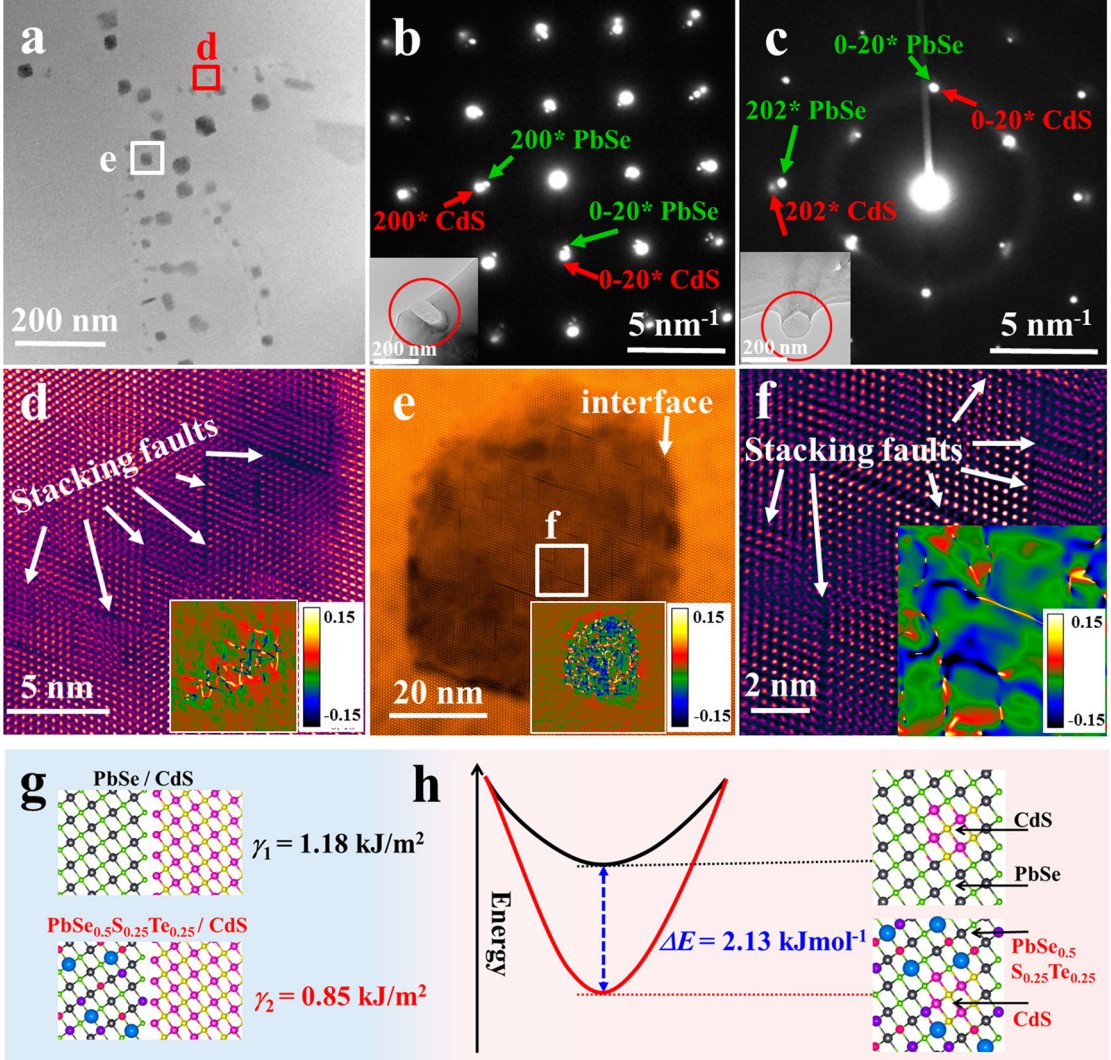

**Fig. 4 Microstructure of Pb$_{0.935}$Cd$_{0.04}$Na$_{0.025}$Se$_{0.5}$S$_{0.25}$Te$_{0.25}$ sample. a** Medium magnification STEM image shows the coexistence of nanoprecipitates and mesoscopic precipitates. Electron diffraction spots along **b** (001) and **c** (110) orientations. The green and red arrows represent spots from PbSe and CdS structures, respectively. **d, e** HAADF images of two typical nanoprecipitates that contain high-density stacking faults. The insets are the GPA results along the $xy$ directions. **f** The enlarged image of the selected area in **e** image. The GPA result shows the tense lattice strains around the stacking faults. **g** Calculated surface energy ($\gamma$) of PbSe/CdS and PbSe$_{0.5}$S$_{0.25}$Te$_{0.25}$/CdS interfaces. **h** Calculated deformation energy of rock salt CdS phase in PbSe and PbSe$_{0.5}$S$_{0.25}$Te$_{0.25}$ lattice. The crystal structures in **g** and **h** represent the calculation model.

should be as small as possible to obtain high surface–volume ratios in nanocomposites for strengthening phonon scattering. The large precipitates will have less of a contribution to reducing lattice thermal conductivity than the small precipitates[17,63]. In this sample, however, the high-density stacking faults within large nanoprecipitates induced dispersed lattice strains not only around interfaces, but also inside the whole particle, which offers strong scattering for heat-carrying phonons.

The high-density stacking faults within nanoprecipitates were interesting and were not reported in previous Cd-doped PbQ-based (Q = S,Se,Te) systems. In previous reports, Cd(S/Se) was stabilized as zinc blende (face-centered cubic) or wurtzite (hexagonal) structure at room temperature, which shows a different atomic arrangement with PbQ (rock salt structure, face-centered cubic)[27,29,30,64]. Therefore, the lattice around the interfaces between Cd(S/Se) and PbQ should be discontinuous. The lattice strains from mismatches were released by the interfaces rather than within the nanoprecipitates. However, upon replacing PbSe by high-entropy PbSe$_{0.5}$S$_{0.25}$Te$_{0.25}$ as a

matrix, the surface energy ($\gamma$) between matrix and nanoprecipitates will be decreased (Fig. 4g). The decreased $\gamma$ will lower the deformation energy of rock salt CdS and stabilized this structure at room temperature (Fig. 4h). The same atomic arrangement of rock salt PbSe$_{0.5}$S$_{0.25}$Te$_{0.25}$ and CdS resulted in a continuous lattice and indistinct interfaces. As a result, the strains from lattice mismatch had to be released by forming stacking faults.

The resulting stacking faults worked as an important role in the phonon-scattering process. Scattering sources from high-entropy composition are usually distributed at the atomic scale (<1 nm)[18] hindering the propagation of high-frequency phonons. However, nano- and meso-precipitates usually scatter low-frequency phonons because of their large size (from a few nanometers to several hundred nanometers)[27,61]. Therefore, the resulting stacking faults with a scale of 1–10 nm provided effective scattering sources for middle-frequency phonons[65,66]. To clarify the contributions of different scattering sources to $\kappa_L$, we calculated the temperature-dependent $\kappa_L$ based on the modified Debye–Callaway model (Fig. 3d). The calculation details can be

found everywhere[23,24,51,65]. We regarded the stacking faults as dislocations because of the similar lattice mismatch. As shown in Fig. 3d, the predicted lines agree well with the experimental data of different samples, which demonstrate that the contributions of different scattering sources should be integrated rather than overlapped. Overall, the hierarchical structure from high-entropy composition, the stacking faults, as well as nano- and meso-precipitates provide effective scattering for full-spectrum phonons (Fig. 1a). These phonons contribute to the very low lattice thermal conductivity.

## Discussion

In conclusion, the TE performance of p-type PbSe was effectively improved by alloying Cd, S, and Te. The band convergence due to the decreased energy offset between light and heavy valence bands benefitted from an enhanced power factor. Furthermore, hierarchical structures including high-entropy composition, stacking faults, as well as nano- and meso-precipitates resulted in very low lattice thermal conductivity. Combined with the optimized electrical and thermal transport properties, a high $zT$ value of 2.0 at 900 K was realized in the $Pb_{0.935}Na_{0.025}Cd_{0.04}Se_{0.5}S_{0.25}Te_{0.25}$ sample. Based on the high-performance chalcogenides in this work and bismuth telluride materials, we fabricated a segmented TE module that had a high experimental conversion efficiency of 12% at $\Delta T = 506$ K. This work promoted PbSe-based materials as a practical TE system to accelerate the development of TE technologies in real applications.

## Methods

**Synthesis**. Pb (shots, 99.99%, Aladdin), Na (shots, 99%, Aladdin), Cd (shots, 99.99%, Aladdin), S (pieces, 99.999%, Aladdin), Se (shots, 99.999%, Aladdin), and Te (pieces, 99.999%, Aladdin) were used to synthesize the ingots in sealed silica tubes under vacuum. The tubes were melted at 1423 K for 7 h before being quenched into cold water, and annealed at 823 K for 2 days. The ground powder was sintered by Spark Plasma Sintering System at 853 K under a pressure of 50 MPa for 10 min.

**Measurements**. The measurement methods can be found in our previous work[51]. The electrical transport properties (electric conductivity and Seebeck coefficient), thermal conductivity, and Hall coefficient were measured by commercial Ulvac ZEM-3, Netzsch LFA 457, and Lake Shore 8400 Series, respectively.

**Calculation**. DFT calculations were performed by the Perdew–Burke–Ernzerhof functional of the generalized gradient approximation, as implemented in the Vienna ab initio Simulation Package.

**TE module**. $Co_{0.8}Fe_{0.2}$ alloy was bonded to the Pb-based materials as barrier layer in hot side. The $Bi_2Te_3$-based materials were electroplated with Ni as barrier layer. The low- and high-temperature legs were connected by Sn–Pb–Ag alloys. Eight pairs of n–p couples were soldered with two direct-bonded copper alumina ceramics by $Sn_{32}Bi_{17}Ag_{51}$ alloys to assemble a segmented TE module. The electrical power output and the conversion efficiency of the segmented module were measured, using the commercial Power Generation Efficiency Characteristics Evaluation System (PEM-2, Advance Riko, Japan).

## Data availability

The authors declare that the data supporting the findings of this study are available within the paper and its Supplementary Information files. All of the other data are available from the authors upon reasonable request.

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

## Acknowledgements

We acknowledge the support from the leading talents of Guangdong Province Program (Grant No. 00201517), Guangdong-Hong Kong-Macao Joint Laboratory (Grant No. 2019B121205001), the National Natural Science Foundation of China (Grant Nos. 52002167, 11874194, 11934007, and 51632005), the Science and Technology Innovation Committee Foundation of Shenzhen (Grant Nos. KQTD2016022619565991, JCYJ20200109141205978 and ZDSYS20141118160434515), and high-level special funds (G02206302). We thank the support from SUSTech Core Research Facilities.

## Author contributions

B.J. and J.H. conceived and designed the experiments; B.J. synthesized the samples and carried out the transport property measurements. Y.Y. and L.X. performed the TEM observations and analysis. H.C. and J.C. performed the calculations. B.J. and X.L. studied the fabrication and characterization of module. B.J. and J.H. wrote, and all of the authors edited this manuscript.

## Competing interests

The authors declare no competing interests.
