## [Peer Review File · Nature Communications]

REVIEWER COMMENTS

Reviewer #1 (Remarks to the Author):

The manuscript entitled 'Entropy engineering promotes thermoelectric performance in p-type PbSe based materials' reports a new strategy to enhance thermoelectric performance of p-type PbQ (Q = chalcogen) based materials. The authors show that Na doping at cationic site can increase the solubility of Te and S at anion sites by raising the matrix entropy. This induces dense point defects, enhances phonon scattering and therefore lowers lattice thermal conductivity. Importantly, they displayed that further Cd doping converged the light and heavy valence band of PbSe and induced abundant precipitates in the matrix. These effects simultaneously increase power factor and reduce the lattice thermal conductivity of PbSe, giving rise to impressively high ZT value of 2.0 at ~900 K. The manuscript contains proper research novelty for Nature Communications. The data are well-organized and the manuscript is well-written. This manuscript is highly recommended for the publication in Nature Communications after properly addressing the following questions.

1. PbSe has been regarded as a promising contender to replace PbTe which containing highly rare and expensive Te. In this work, the title material is $\text{Pb}_{0.975}\text{Na}_{0.025}\text{Se}_{0.5}\text{S}_{0.25}\text{Te}_{0.25}$. It contains 25% of Te! Thus, it is very strange to claim this material as PbSe-based materials given typical so-called PbSe-based systems contains at most a few % doping and alloying elements. It is well established that the presence of Te greatly enhances power factor and reduces thermal conductivity. As a result, it is unfair to compare thermoelectric properties of 25% Te alloyed systems with those of other PbSe-based materials. Thus, the term "PbSe-based" should not be used.
2. The authors need to plot the lattice parameter with respect to doping concentration. It is hard for readers to visualize the relationship between doping concentration and lattice parameters.
3. The authors provide nice image about the stacking faults within the Cd-rich precipitates. However, precipitates scatter phonons depending mainly on its size and interface mismatch to the surrounding matrix. Accordingly, please clarify how these stacking faults in the precipitates benefit the phonon scattering. Explanation based on well-defined models such as Debye-Callaway model will be helpful.
4. The authors calculated the band structures with a $3 \times 3 \times 3$ supercell. Which was Cd placed for the calculation, nearby Se or surrounded by Te and S? Is there any difference in band structures according to the different Cd site in different supercell models? The authors need to clearly define the supercell in the supporting information otherwise it is hard for readers to reproduce the same results.
5. Please provide Pisarenko relationship to reflect the effect of band engineering.
6. The hall coefficient for Cd doped sample peaks at nearly the same temperature, indicating Cd marginally converged the valence bands. This seems contrary to their calculation shown in Figure S1a. Please explain.
7. Despite the authors focused on the p-type PbSe, this reviewer recommend them to properly cite recently published high-performance n-type PbSe materials. This deepens readers' understanding about PbSe based thermoelectric materials.

The present manuscript provides remarkable findings concerning the performance of the thermoelectric PbSe upon doping. This is a timely topic since PbSe based thermoelectrics are very promising materials for power generation and cooling. The present investigation demonstrates how a sophisticated doping strategy can tailor both the electrical conductivity and the Seebeck coefficient and reach a peak zT of 2.0 at 900 K. This is a noteworthy accomplishment since PbSe becomes an economic alternative to PbTe with such a quality factor. The authors attribute the low thermal conductivity to entropy engineering. It is this last statement, or on more general grounds the question how this excellent thermoelectric performance can be understood, where the manuscript should be extended. Doing this could help to advance the art of creating superior thermoelectric materials.

Specifically, the authors should explain how Cd doping leads to better band convergence and hence an enhanced Seebeck coefficient and good electric transport properties. Recently, the band structure of lead chalcogenides has been attributed to the crucial role of p-electrons forming a σ -bond [Paper Snyder (Chemistry of Materials 32, 9771 (2020)]. This configuration, which is very different from ordinary covalent semiconductors has been attributed to metavalent bonding [Advanced Materials 30, 1803777 (2018), Advanced Materials 32, 1908302 (2020)]. Indeed, a recent paper employs this bonding mechanism to explain the superior thermoelectric performance of doped GeSe to this bonding mechanism [Angewandte Chemie International Edition, doi 10.1002/anie.202101283]. The same group also has argued recently, that Cd doping of AgSbTe₂ attributed to improved ordering! [Science 371, 722 – 727 (2021)], while the present manuscript argues in striking contrast that increasing disorder, i.e. entropy is responsible for the superior performance. These two different views should be discussed, in particular since no compelling evidence is presented here that Cd doped PbSe and its alloys are truly governed by entropic effects and that metavalent bonding can be excluded as an explanation for the observations reported here. There is instead strong evidence that metavalent bonding governs the properties of lead mono-chalcogenides such as PbSe [Advanced Materials 32, 202005533 (2020)].

At this point in time it is still premature to argue which of the two views is correct. Yet, there is evidence that the superior performance of Cd doped PbSe is not due to entropy engineering. In particular, it seems possible that metavalent bonding and its modification by Cd doping lead to superior thermoelectric performance. This view should also be discussed in the manuscript, in particular since it has recently been demonstrated how MVB can explained the superior thermoelectric performance of mono-chalcogenides [Advanced Materials 30, 1801787 (2018); Advanced Functional Materials 29, 1904862 (2019); Angewandte Chemie International Edition, doi 10.1002/anie.202101283].

Reviewer #3 (Remarks to the Author):

This paper reports the thermoelectric properties of a PbSe-based high-entropy alloy. The entropy-stabilized composition is shown to have impressive thermoelectric properties, which the authors attribute to the synergistic effects of band convergence and multi-scale phonon scattering. The claims are well supported by extensive characterizations (spectroscopy, transport, and electron microscopy) and simulations. The achieved thermoelectric ZT is good and an energy conversion device with 12% efficiency is also demonstrated. This study is very thorough and solid, and all conclusions are well supported by the data. There is no doubt that this manuscript should be published in some form somewhere. My main objection to its publication in Nature Communications is its lack of novelty. The mechanisms discussed in this work, including entropy-stabilized alloy, band convergence, and multi-scale phonon scattering, have all been extensively explored before and well understood in the thermoelectric community. The material system itself is also a classic system without many surprises or new physical insights. Thus, I cannot recommend its publication in Nature Communications and would recommend resubmission to a more specialized journal.

Reviewer: 1

General comments: The manuscript entitled ‘Entropy engineering promotes thermoelectric performance in p-type PbSe based materials’ reports a new strategy to enhance thermoelectric performance of p-type PbQ (Q = chalcogen) based materials. The authors show that Na doping at cationic site can increase the solubility of Te and S at anion sites by raising the matrix entropy. This induces dense point defects, enhances phonon scattering and therefore lowers lattice thermal conductivity. Importantly, they displayed that further Cd doping converged the light and heavy valence band of PbSe and induced abundant precipitates in the matrix. These effects simultaneously increase power factor and reduce the lattice thermal conductivity of PbSe, giving rise to impressively high ZT value of 2.0 at ~900 K. The manuscript contains proper research novelty for Nature Communications. The data are well-organized and the manuscript is well-written. This manuscript is highly recommended for the publication in Nature Communications after properly addressing the following questions.

Response: We would like to thank reviewer 1 for his/her much appreciating our work, and for the comments and suggestions as well. We have carefully taken the comments into consideration in preparing our revision.

Comment 1. PbSe has been regarded as a promising contender to replace PbTe which containing highly rare and expensive Te. In this work, the title material is $\text{Pb}_{0.975}\text{Na}_{0.025}\text{Se}_{0.5}\text{S}_{0.25}\text{Te}_{0.25}$. It contains 25% of Te! Thus, it is very strange to claim this material as PbSe-based materials given typical so-called PbSe-based systems contains at most a few % doping and alloying elements. It is well established that the presence of Te greatly enhances power factor and reduces thermal conductivity. As a result, it is unfair to compare thermoelectric properties of 25% Te alloyed systems with those of other PbSe-based materials. Thus, the term “PbSe-based” should not be used.

Response: Thanks for this suggestion. We have changed the term “PbSe-based materials” into “**chalcogenides**”.

Comment 2. The authors need to plot the lattice parameter with respect to doping concentration. It is hard for readers to visualize the relationship between doping concentration and lattice parameters.

Response: Good point. We have plotted the lattice parameter with respect to doping concentration (Fig. R1, Fig. 2a in the main text). As the figure shows, the lattice parameter decreased with increasing Cd content.

Fig. R1 Powder XRD patterns (left) and calculated lattice parameters (right) for $\text{Pb}_{0.975-x}\text{Cd}_x\text{Na}_{0.025}\text{Se}_{0.5}\text{S}_{0.25}\text{Te}_{0.25}$ ($x = 0, 0.01, 0.02, 0.03, 0.04, 0.05$) samples.

Comment 3. The authors provide nice image about the stacking faults within the Cd-rich precipitates. However, precipitates scatter phonons depending mainly on its size and interface mismatch to the surrounding matrix. Accordingly, please clarify how these stacking faults in the precipitates benefit the phonon scattering. Explanation based on well-defined models such as Debye-Callaway model will be helpful.

Response: The phonon scattering around interfaces should be related to the size of precipitates, interface mismatch and phonon frequency. Traditionally, only mid- to long-wavelength phonons are effectively scattered by interfaces in nanocomposites (Gang Chen et al. Energy Environ. Sci., 2009, 2, 466-479.). **The phonons whose mean free path is lower than the size of precipitates (low- to mid-wavelength phonons) still transmit across the interfaces and propagate heat** (D. G. Cahill and G. Chen et al. Appl. Phys. Rev., 2014, 1, 011305). In our experiment, the stacking faults inside precipitates have smaller

scales than that of the precipitates. As a result, these low- to mid-wavelength phonons which transmit across the interfaces will be effectively scattered by the stacking faults. We also used widely accepted Debye-Callaway model to evaluate the effect of precipitates and the stacking faults, as Fig. R2 (Fig. 3d in the main text) shows.

We have added a paragraph on page 9 to explain the analysis based on Debye-Callaway model. *“To clarify the contributions of different scattering sources to κ_L , we calculated the temperature-dependent κ_L based on the modified Debye-Callaway model (Fig. 3d). The calculation details can be found everywhere^{23,24,40,65}. We regarded the stacking faults as dislocations because of the similar lattice mismatch. As shown in Fig. 3d, the predicted lines agree well with the experimental data of different samples, which demonstrate that the contributions of different scattering sources should be integrated rather than overlapped.”*

Fig. R2 Lattice thermal conductivity κ_L of $\text{Pb}_{0.975}\text{Na}_{0.025}\text{Se}$, $\text{Pb}_{0.935}\text{Na}_{0.025}\text{Cd}_{0.04}\text{Se}$, and $\text{Pb}_{0.975-x}\text{Cd}_x\text{Na}_{0.025}\text{Se}_{0.5}\text{S}_{0.25}\text{Te}_{0.25}$ ($x = 0, 0.01, 0.02, 0.03, 0.04, 0.05$) samples. The dash lines were calculated by modified Debye-Callaway model. U, NP, PD and SF represent Umklapp process, nanoprecipitates, point defects, and stacking faults scattering for phonons, respectively.

Comment 4. The authors calculated the band structures with a 3*3*3 supercell. Which was Cd placed for the calculation, nearby Se or surrounded by Te and S? Is there any difference in band structures according to the different Cd site in different supercell models? The authors

need to clearly define the supercell in the supporting information otherwise it is hard for readers to reproduce the same results.

Response: We now added Fig. R3 (Fig.S2 in supporting information) to show the supercells in our calculation of band structure. As shown in Fig. R3a-c, we have calculated three supercells. It is seen that Cd is more effective in widening the band gap and promoting the band convergence in $\text{Pb}_{26}\text{CdSe}_{13}\text{Te}_7\text{S}_7-1$ (a) and $\text{Pb}_{26}\text{CdSe}_{13}\text{Te}_7\text{S}_7-2$ (b) when it is bonded with several Se atoms. **Considering the content of anion site, we adopt the result of $\text{Pb}_{26}\text{CdSe}_{13}\text{Te}_7\text{S}_7-1$ (a) finally in the main text, and it is also the best special quasi-random structures (SQS) generated by the alloy theoretic automated toolkit (ATAT).**

Fig. R3 (a)-(c): $\text{Pb}_{26}\text{CdSe}_{13}\text{Te}_7\text{S}_7$ with Cd at different positions, marked as $\text{Pb}_{26}\text{CdSe}_{13}\text{Te}_7\text{S}_7-1$, $\text{Pb}_{26}\text{CdSe}_{13}\text{Te}_7\text{S}_7-2$ and $\text{Pb}_{26}\text{CdSe}_{13}\text{Te}_7\text{S}_7-3$, respectively. Cd atom is surrounded by (a) 3 Se atoms, 2 S atoms and 1 Te atom; (b) 5 Se atoms and 1 S atom; and (c) 1 Se atom, 2 S atoms and 3 Te atoms. (d) The band structures of Cd doped high-entropy composition and the high-entropy composition without Cd.

Comment 5. Please provide Pisarenko relationship to reflect the effect of band engineering.

Response: We have plotted the Pisarenko relationship to reflect the effect of band

engineering, as Fig. R4 (Fig. 2f in the main text) shows. We can see that the Seebeck coefficient is obviously larger than the theoretical line, demonstrating the increased carrier effective mass and band convergence.

Fig. R4 Carrier concentration dependence of the Seebeck coefficients (the Pisarenko relationship) for $\text{Pb}_{0.975-x}\text{Cd}_x\text{Na}_{0.025}\text{Se}_{0.5}\text{S}_{0.25}\text{Te}_{0.25}$ ($x = 0, 0.01, 0.02, 0.03, 0.04, 0.05$) samples. The black line and open symbols are from reference²⁷.

Comment 6. The hall coefficient for Cd doped sample peaks at nearly the same temperature, indicating Cd marginally converged the valence bands. This seems contrary to their calculation shown in Figure S1a. Please explain.

Response: As Fig. R5a (Fig. 2e in the main text) shows, **the temperature of hall coefficient peak is decreased from 600 K ($x = 0$ sample) to 500K ($x = 0.02$ sample)**. This phenomenon is similar to the reported results in Hg-doped PbSe and Cd-doped Pb(Se,Te). In Hg-doped PbSe, the peak temperature is decreased from 650 K ($x = 0$ sample) to 550 K ($x = 0.02$ sample) (M. G. Kanatzidis, J. Am. Chem. Soc., 2018, 140, 18115-18123.). In Cd-doped Pb(Se,Te), the peak temperature is also decreased from 650 K ($x = 0$ sample) to 550 K ($x = 0.03$ sample) (M. G. Kanatzidis, J. Am. Chem. Soc., 2019, 141, 4480-4486.).

With further increasing Cd content, the peak temperature will be similar. The reason is that the solubility of Cd element is lower than 0.02. The introduced Cd over solubility limit in $x \geq 0.02$ samples should enter the nanoprecipitates rather than the matrix. Thus the band structure should be less affected. L. D. Zhao also reported this phenomenon in Cd-doped PbSe, as Fig. R5b shows (L. D. Zhao et al. J. Am. Chem. Soc., 2013, 135, 7364-7370.).

[Redacted]

Fig. R5 (a) Temperature dependence of Hall coefficients for $\text{Pb}_{0.975-x}\text{Cd}_x\text{Na}_{0.025}\text{Se}_{0.5}\text{S}_{0.25}\text{Te}_{0.25}$ ($x = 0, 0.01, 0.02, 0.03, 0.04, 0.05$) samples. (b) Band energy differences as a function of Cd fractions in PbSe for the conduction C , light hole L , and heavy hole Σ bands (L. D. Zhao et al. J. Am. Chem. Soc., 2013, 135, 7364-7370.).

Comment 7. Despite the authors focused on the p-type PbSe, this reviewer recommend them to properly cite recently published high-performance n-type PbSe materials. This deeps readers' understanding about PbSe based thermoelectric materials.

Response: We have added the recent publications about n-type PbSe materials in introduction section.

Reviewer: 2

General comments: The present manuscript provides remarkable findings concerning the performance of the thermoelectric PbSe upon doping. This is a timely topic since PbSe based thermoelectrics are very promising materials for power generation and cooling. The present investigation demonstrates how a sophisticated doping strategy can tailor both the electrical conductivity and the Seebeck coefficient and reach a peak zT of 2.0 at 900 K. This is a noteworthy accomplishment since PbSe becomes an economic alternative to PbTe with such a quality factor.

Response: We would like to Reviewer 2 for his/her thoughtful review of our manuscript. We have carefully taken the comments into consideration in our revised manuscript.

Comments: The authors attribute the low thermal conductivity to entropy engineering. It is this last statement, or on more general grounds the question how this excellent thermoelectric performance can be understood, where the manuscript should be extended. Doing this could help to advance the art of creating superior thermoelectric materials.

Specifically, the authors should explain how Cd doping leads to better band convergence and hence an enhanced Seebeck coefficient and good electric transport properties. Recently, the band structure of lead chalcogenides has been attributed to the crucial role of p-electrons forming a σ -bond [Paper Snyder (Chemistry of Materials 32, 9771 (2020)]. This configuration, which is very different from ordinary covalent semiconductors has been attributed to metavalent bonding [Advanced Materials 30, 1803777 (2018), Advanced Materials 32, 1908302 (2020)]. Indeed, a recent paper employs this bonding mechanism to explain the superior thermoelectric performance of doped GeSe to this bonding mechanism [Angewandte Chemie International Edition, doi 10.1002/anie.202101283] . The same group also has argued recently, that Cd doping of AgSbTe₂ attributed to improved ordering! [Science 371, 722 – 727 (2021)], while the present manuscript argues in striking contrast that increasing disorder, i.e. entropy is responsible for the superior performance. These two different views should be discussed, in particular since no compelling evidence is presented here that Cd doped PbSe and its alloys are truly governed by entropic effects and that metavalent bonding can be excluded as an explanation for the observations reported here. There is instead strong evidence that metavalent bonding governs the properties of lead mono-chalcogenides such as PbSe [Advanced Materials 32, 202005533 (2020)].

At this point in time it is still premature to argue which of the two views is correct. Yet, there is evidence that the superior performance of Cd doped PbSe is not due to entropy engineering. In particular, it seems possible that metavalent bonding and its modification by Cd doping lead to superior thermoelectric performance. This view should also be discussed in the manuscript, in particular since it has recently been demonstrated how MVB can explained the superior thermoelectric performance of mono-chalcogenides [Advanced Materials 30, 1801787 (2018); Advanced Functional Materials 29, 1904862 (2019); Angewandte Chemie International Edition, doi 10.1002/anie.202101283].

Response: The reviewer 2 raised a very good point. We believe that the MVB theory can help us further understand the electrical and thermal transport properties in chalcogenides. We have made some changes in our revised manuscript, as described below:

1. The band structure of chalcogenides (PbS, PbSe, PbTe) is closely related to the σ -bonding configuration and orbital hybridization. Thus **the changed electrical transport properties should also be affected by the metavalent bonding**. We have added a paragraph on page 4 in the main text to analyze the effect of changed metavalent bonding because of alloyed Cd on electrical transport properties. The paragraph is “*PbSe is well known as a typical incipient metal with a unique bonding mechanism called metavalent bonding (MVB)⁵¹. MVB is mainly formed by p-orbitals in a σ -bonding configuration in chalcogenides, which shows high electron mobility because of the small conductivity effective mass and weak s-p hybridization⁵². In our experiment, alloying Cd at Pb site will strengthen s-p hybridization between cation and anion because of the increased charge sharing^{51,53}. Thus, the band gap opens and band effective mass of a single valley (m_b^*) increases, resulting in the reduced charge carrier mobility (Fig. S1d)⁵⁴. In this regard, alloying Cd should deteriorate the electrical transport property based on a single parabolic band model. However, the enlarged band gap decreases the energy separation between L and Σ bands and promotes the band convergence as verified by our DFT calculations. The participation of Σ band in electrical transport process leads to the multiple band behavior¹³. As a consequence, the valley degeneracy N_v and density-of-states effective mass m^* ($m^* = N_v^{2/3} m_b^*$) will be largely increased (the Pisarenko line in Fig. 2f)¹³, resulting in the enhanced S and power factor ($PF = S^2 \sigma$).”*
2. **We trust that the thermal transport properties should also be largely affected at the heterointerface (CdS and PbSe interface) because of the distinctively different chemical bonding mechanisms between covalent CdS and MVB PbSe.** We also added a paragraph on page 7 in the main text to explain this effect on lattice thermal conductivity. The paragraph is “*Generally, CdS exhibits covalent bonding, which shows different physical properties, such as optical dielectric constant, effective coordination numbers and Born effective charge compared with MVB materials Pb(Se,S,Te)⁵¹⁻⁵³. Thus,*

there should be large phonon scattering at the MVB/covalent heterointerfaces due to the very different spring constant of chemical bonds, contributing to the ultralow κ_L ⁶⁰.”.

3. To further explain the effect of entropy engineering on electrical and thermal transport properties, we added two paragraphs in the main text.

On page 3 in the main text, we explained the increased atomic ordering because of entropy-driven structural stabilization. **The increased entropy will change multiphase composition to a single phase, resulting in the stabilized structure and enhanced atomic ordering**, which is similar to the improved atomic ordering in Cd-doped AgSbTe₂ (Science 371, 722 – 727 (2021)). The paragraph on page 3 is *“The stabilized structure can keep the long-range order of atomic arrangement, thereby eliminating the boundary phonon scattering around phase interfaces¹⁸. This phenomenon of stabilized structure from entropy-driven structural stabilization can maintain the electrical transport framework and improve the electrical properties¹⁸, which is similar to the improved atomic ordering in Cd-doped AgSbTe₂⁵⁰. Meanwhile, the increased solubility of Te and S at anion sites from the increased entropy also extends phase space for performance optimization and largely distorts the lattice, resulting in strong scattering for heat-carrying phonons.”.*

We further explained the effect of entropy engineering on thermal transport properties. The strong lattice strain from distorted lattice has been observed and its effect on lattice thermal conductivity was also analyzed in our previous paper (Science, 371, 830-834 (2021)). In this communication, **we emphasized the effect of entropy engineering on hierarchical structure. Because of the changed surface energy from entropy engineering, the interface between nanoprecipitates and high-entropy matrix was tuned, resulting in high-density stacking faults inside the nanoprecipitates.** The scale of the stacking faults is between point defects and nanoprecipitates. As a result, the full-spectrum phonons should be scattered by the hierarchical structure, resulting in the ultralow lattice thermal conductivity. We also used modified Debye-Callaway model to illustrate the effect of this hierarchical structure formed by entropy engineering. The added paragraph on page 9 is *“To clarify the contributions of different scattering sources*

to κ_L , we calculated the temperature-dependent κ_L based on the modified Debye-Callaway model (Fig. 3d). The calculation details can be found everywhere^{23,24,40,65}. We regarded the stacking faults as dislocations because of the similar lattice mismatch. As shown in Fig. 3d, the predicted lines agree well with the experimental data of different samples, which demonstrate that the contributions of different scattering sources should be integrated rather than overlapped.”

Reviewer: 3

General comments: This paper reports the thermoelectric properties of a PbSe-based high-entropy alloy. The entropy-stabilized composition is shown to have impressive thermoelectric properties, which the authors attribute to the synergistic effects of band convergence and multi-scale phonon scattering. The claims are well supported by extensive characterizations (spectroscopy, transport, and electron microscopy) and simulations. The achieved thermoelectric ZT is good and an energy conversion device with 12% efficiency is also demonstrated. This study is very thorough and solid, and all conclusions are well supported by the data. There is no doubt that this manuscript should be published in some form somewhere. My main objection to its publication in Nature Communications is its lack of novelty. The mechanisms discussed in this work, including entropy-stabilized alloy, band convergence, and multi-scale phonon scattering, have all been extensively explored before and well understood in the thermoelectric community. The material system itself is also a classic system without many surprises or new physical insights. Thus, I cannot recommend its publication in Nature Communications and would recommend resubmission to a more specialized journal.

Response: We thank the reviewer 3 for appreciating our work and critical comment on novelty. We would like to say that our work for the first time presented **the synergistic effect of entropy engineering, band convergence and hierarchical structure**, which is novel and significant in high-entropy thermoelectric materials. Some explanations are summarized below:

Firstly, the core conclusion in this manuscript is that entropy engineering can work together with traditional optimization methods (band convergence and hierarchical structure) to further improve thermoelectric performance. Although many methods have been proposed to improve thermoelectric performance, the combination of these methods won't definitely result in synergistic effect. The reason is that the effects of some methods are similar or opposite. For example, there are no reports about the combination of band convergence and resonant level because of the similar effect of the increased density-of-states effective mass. Nanostructure is beneficial to reduce lattice thermal conductivity, but often results in largely deteriorated electrical transport properties and decreased zT values. Thus it is very meaningful to explore the possible synergistic effect of different optimization mechanisms. For example, Kanishka Biswas et al. proved that hierarchical structure composed of point defects, nanoscale and mesoscale precipitates can work together to scatter full-spectrum phonons and resulted in ultralow lattice thermal conductivity (Nature, 489, 414-418 (2012)). So it is the same with this work. We found that **entropy-stabilized composition provides extended composition space which can simultaneously introduce band convergence and hierarchical structure for optimizing property**. These three methods (entropy-stabilized composition, band convergence, hierarchical structure) can synergistically optimize electrical and thermal transport properties, resulting in a **step-by-step improvement of zT value** (Fig. R6).

Fig. R6 zT values of $\text{Pb}_{0.975}\text{Na}_{0.025}\text{Se}$, $\text{Pb}_{0.935}\text{Na}_{0.025}\text{Cd}_{0.04}\text{Se}$, and entropy-stabilized $\text{Pb}_{0.975-x}\text{Cd}_x\text{Na}_{0.025}\text{Se}_{0.5}\text{S}_{0.25}\text{Te}_{0.25}$ ($x = 0, 0.01, 0.02, 0.03, 0.04, 0.05$) samples.

Secondly, **we also found that hierarchical structure can be produced by entropy engineering**, which was not reported in previous literature. Using high-entropy composition as matrix, the surface energy of CdS nanoprecipitates can be decreased and face-centered-cubic structure was stabilized to room temperature. The stabilized structure of CdS resulted in high-density stacking faults inside the precipitates. Thus **we obtained an entire hierarchical structure composed of high-entropy composition, the stacking faults, nanoscale and mesoscale precipitates**, which should be more effective for phonon scattering than the traditional one without high-entropy composition and the stacking faults. Finally, this work realized a high experimental conversion efficiency of 12% in PbSe-based segmented thermoelectric module. This low-cost module even showed higher conversion efficiency than the widely studied PbTe and Skutterudites system. **The largely enhanced conversion efficiency and decreased material cost should promote real applications of thermoelectric technology.** Therefore, this work will appeal to the attentions of the thermoelectric community.

REVIEWERS' COMMENTS

Reviewer #1 (Remarks to the Author):

The revised manuscript has been greatly improved, and can be published in Nature Communications. The manuscript is written well, and the science in this work is novel enough to be published in this esteemed journal.

Thermoelectric performance reported in this work is undoubtedly superior to other competing systems.

Reviewer #2 (Remarks to the Author):

The authors have carefully considered the comments of all referees. I am satisfied by all changes implemented and the revised presentation and discussion of the data. Since the comments of all referees have been considered thoroughly, my recommendation would be to publish the paper in its present state.

Reviewer #3 (Remarks to the Author):

I thank the authors for their efforts in addressing my previous comments in great detail. Their main argument is that although the three mechanisms (entropy-stabilized alloy, band convergence, and hierarchical phonon scattering) are already well established, realizing the synergistic effect of the three mechanisms is novel and nontrivial. I partially agree with this argument in that it is not straightforward to combine these three mechanisms in a single practical material system; however, the impact of this work is limited since there are no new concepts proposed and no general guidelines provided on how to implement these synergistic effects. On the other hand, I do agree with the other two reviewers that the achieved zT and power conversion performance are very good based on the PbSe material, although the discovered material composition is quite specific. In this case, I will leave the decision to the editor on whether this work reporting good thermoelectric performance but lacking fundamental novelty can be published in Nature Communications.

Reviewer: 1

General comments: The revised manuscript has been greatly improved, and can be published in Nature Communications. The manuscript is written well, and the science in this work is novel enough to be published in this esteemed journal. Thermoelectric performance reported in this work is undoubtedly superior to other competing systems.

Response: We would like to thank reviewer 1 for his/her much appreciating our work again, and highly recommendation for publication in Nature communications.

Reviewer: 2

General comments: The authors have carefully considered the comments of all referees. I am satisfied by all changes implemented and the revised presentation and discussion of the data. Since the comments of all referees have been considered thoroughly, my recommendation would be to publish the paper in its present state.

Response: We thank the reviewer 2 once more for the very positive comments and recommendation.

Reviewer: 3

General comments: I thank the authors for their efforts in addressing my previous comments in great detail. Their main argument is that although the three mechanisms (entropy-stabilized alloy, band convergence, and hierarchical phonon scattering) are already well established, realizing the synergistic effect of the three mechanisms is novel and nontrivial. I partially agree with this argument in that it is not straightforward to combine these three mechanisms in a single practical material system; however, the impact of this work is limited since there are no new concepts proposed and no general guidelines provided on how to implement these synergistic effects. On the other hand, I do agree with the other two reviewers that the achieved zT and power conversion performance are very good based on the PbSe material, although the discovered material composition is quite specific. In this case, I will leave the decision to the editor on whether this work reporting good thermoelectric performance but lacking fundamental novelty can be published in Nature Communications.

Response: We would like to thank the reviewer 3 again for carefully assessing the manuscript. As his/her wrote, this work first reported good thermoelectric performance in low-cost p-type PbSe-based system, including high zT value in material and high conversion efficiency in module. **This result shows important significance in thermoelectric community, which should promote the real applications of thermoelectric technology.** Secondly, **this work provides a guideline for similar thermoelectric materials, such as GeTe, SnTe.** These materials show the same crystal structure to PbSe, demonstrating the possibility of forming high-entropy-stabilized composition. In addition, their thermoelectric properties also have been improved by band convergence and hierarchical structure. Thus the synergistic effect of high-entropy-stabilized composition, band convergence and hierarchical structure presented in this work should provide more possibility for traditional thermoelectric materials.